# How Transformers Reason: A Case Study on a Synthetic Propositional Logic Problem

**Guan Zhe Hong**[*][1]    **Nishanth Dikkala** [2]    **Enming Luo** [2]    **Cyrus Rashtchian** [2]
**Xin Wang** [2]    **Rina Panigrahy** [2]
[1]Purdue University    [2] Google Research
hong288@purdue.edu,
{nishanthd, enming, cyroid, wanxin, rinap}@google.com

## Abstract

Large language models (LLMs) have demonstrated remarkable performance in tasks that require reasoning abilities. Motivated by recent works showing evidence of LLMs being able to plan and reason on abstract reasoning problems in context, we conduct a set of controlled experiments on a synthetic propositional logic problem to provide a mechanistic understanding of how such abilities arise. In particular, for a decoder-only Transformer trained solely on our synthetic dataset, we identify the specific mechanisms by which a three-layer Transformer solves the reasoning task. In particular, we identify certain "planning" and "reasoning" circuits which require cooperation between the attention blocks to in totality implement the desired reasoning algorithm. To expand our findings, we then study a larger model, Mistral 7B. Using activation patching, we characterize internal components that are critical in solving our logic problem. Overall, our work systemically uncovers novel aspects of small and large transformers, and continues the study of how they plan and reason.

## 1 Introduction

Language models using the transformer architecture [Vaswani et al., 2017] have shown remarkable capabilities on many natural language tasks [Brown et al., 2020, Radford et al., 2019b]. Trained with causal language modeling wherein the goal is next-token prediction on huge amounts of text, these models exhibit deep language understanding and generation skills. An essential milestone in the pursuit of models which can achieve a human-like artificial intelligence, is the ability to perform human-like reasoning and planning in complex unseen scenarios. While some recent works using probing analyses have shown that the activations of the deeper layers of a transformer contain rich information about certain mathematical reasoning problems [Ye et al., 2024], the question of what mechanisms inside the model enables such abilities remains unclear.

While the study of how transformers reason in general remains a daunting task, in this work, we aim to improve our *mechanistic* understanding of how a Transformer reason through simple propositional logic problems. For concreteness' sake, consider the following problem:

Rules: A or B implies C. D implies E. Facts: A is true. B is false. D is true.
Question: what is the truth value of C?

An answer with *minimal* proof is "A is true. A or B implies C; C is true."

The reasoning problem, while simple-looking on the surface, requires the model to perform several actions that are essential to more complex reasoning problems, all without chain of thought (CoT).

---

[*]Work done as a student researcher at Google Research.

Before writing down any token, the model has to first discern the *rule* which is being queried: in this case, it is "A or B implies C". Then, it needs to rely on the premise variables A and B to the locate the relevant *facts*, and find "A is true" and "B is false". Finally, it needs to decide that "A is true" is the correct one to invoke in its answer due to the nature of disjunction. It follows that, to write down the first token "A", the model already has to form a "mental map" of the variable relations, value assignments and query! Therefore, we believe that this is close to the minimal problem to examine how a model internalizes and plans for solving a nontrivial mathematical reasoning problem where apparent ambiguities in the problem specification cannot be resolved trivially.

To understand the internal mechanisms of how a transformer solves problems resembling the minimal form above, we perform two flavors of experiments. The first is on shallow transformers trained purely on the synthetic propositional logic problems. This enables a fine-grained analysis in a controlled setting. The other set of experiments are on a pre-trained LLM (Mistral-7B), where we primarily rely on activation patching to uncover necessary circuits for solving the reasoning problem, including specialized roles of certain components. At a high level, we make the following discoveries based on our two fronts of analysis:

1. We discover that small transformers, trained purely on the synthetic problem, utilize certain "*routing embeddings*" to significantly alter the information flow of the deeper layers when solving different sub-categories of the reasoning problem. We also characterize the different reasoning pathways: we find that problems querying for reasoning chains involving logical operators typically require greater involvement of all the layers in the model.

2. We uncover properties of the circuit which the pretrained LLM Mistral-7B-v0.1 employs to solve the minimal version of the reasoning problem. We find four families of attention heads, which have surprisingly specialized roles in processing different sections of the context: queried-rule locating heads, queried-rule mover heads, fact-processing heads, and decision heads. We find evidence suggesting that the model follows the natural reasoning path of "QUERY→Relevant Rule→Relevant Fact(s)→Decision".

We discuss related works and scope of this work in detail in Appendix A.

## 2 Problem setting

In this section, we present the core properties of the synthetic propositional logic problem which shall be the data model of this paper. We delay finer details and more examples of the problem to Appendix B.

### 2.1 Data model: a propositional logic problem

Our problem follows an implicit causal structure, as illustrated in Figure 1. The structure consists of two distinct chains: One containing a logical operator at the end of the chain, and the other forming a purely linear chain.

We require the model to generate a *minimal* reasoning chain, consisting of "relevant facts", proper rule invocations, and intermediate truth values, to answer the truth-value query. Consider an example constructed from the causal graph in Figure 1, written in English:

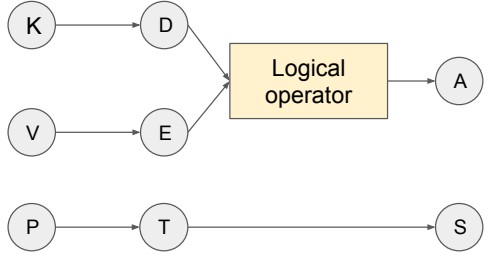

- Rules: K implies D. D or E implies A. V implies E. T implies S. P implies T.
- Facts: K is true. P is true. V is false.
- Query: A.
- Answer: K is true. K implies D; D is true. D or E implies A; A is true.

In this example, the QUERY token A is the terminating node of the OR chain. Since any *true* input to an OR gate (either D or E) results in

Figure 1: Synthetic data model. The causal structure has two chains: one with a logical operator (LogOp) at the end and the other being purely a linear causal chain. This example is the length-3 case.

A being *true*, the minimal solution chooses only one of the starting nodes from the OR chain to construct its argument: in this case, node K is chosen.

# 3 Mechanisms of planning and reasoning: a case study of the length-3 problem

In this section, we discuss the internal mechanisms of a small transformer trained purely on the synthetic problem. While there are many parts of the answer of the transformer which can lead to interesting observations, due to space limitations, we primarily focus on the model's "mental process" for producing the most important part of the answer, namely the *first token*. To further justify this choice, we find that on our problem, a model's full-answer accuracy strongly correlates with its accuracy of the first answer token, as detailed in Figure 4 in the Appendix.

**Architecture choice**. We study a decoder-only attention-only transformer closely resembling the form of GPT-2 [Radford et al., 2019a]. We discuss training and architecture details in the Appendix. We select the smallest transformer that can achieve 100% accuracy (or sufficiently close to it) to initiate our analysis, a 3-layer 3-head variant.

## 3.1 Empirical observations

We begin our analysis with mechanisms that are universal to how the model plans and reasons for predicting the first token. Then we describe the mechanisms that only arise when the model needs to deal with specific situations. We discuss the main observations here, and leave the quantitative details to Appendix D.

**Mental notes at the QUERY position**. The QUERY token is likely the most important token in the context: it determines which chain is being queried. The transformer makes use of this token in its answer in an intriguing way.

*Observation 1: chain-type disentanglement at QUERY*. We observe that, the second layer's attention block exhibit disentanglement in its output direction dependent on whether it is the linear chain that is being queried. Intriguingly, the third layer's attention heads place greater than 90% of their attention weights on the QUERY position on average when the linear chain is queried.

Based on Observation 1, we hypothesize that given a chain type (linear or LogOp), there exists certain directions at the second attention block which somehow change the behavior of the third attention block: attracting its attention to QUERY when it is the linear chain, and pushing its attention away from QUERY when it is the LogOp chain. We confirm the existence and role of this "routing" signal.

*Observation 2: existence of an abstract "routing signal"*. We compute the average of the second attention block's output on 1k samples whose QUERY is for the linear chain, which we denote as $h_{route}$. There are two interesting properties of this embedding direction:

1. (Linear→LogOp intervention) We generate 500 test samples where QUERY is for the linear chain. *Subtracting* the embedding $h_{route}$ from the second attention block's output results in the model outputting the correct first token for the *LogOp chain* of the problem 100% of the time on the test samples. In other words, the "mode" in which the model reasons is flipped from "linear" to "LogOp".
2. (LogOp→linear intervention) We generate 500 test samples where QUERY is for the LogOp chain. *Adding* $h_{route}$ to the second attention block's output causes the three attention heads in layer 3 to focus on the QUERY position: greater than 99% of the attention weights are on this position averaged over the test samples. In this case, however, the model does not output the correct starting node for the linear chain on more than 90% of the test samples.

It follows that there indeed exists an *abstract embedding direction* inside the transformer which *significantly changes the information flow depending on the chain type being queried*.

**Linear chain**. At this point, it is clear to us that, when QUERY is for the linear chain, the third layer mainly serves a simple "message passing" role at the QUERY position. A natural question arises: does the input to the third layer truly contain the information to determine the first token of the answer, namely the starting node of the linear chain? The answer is yes.

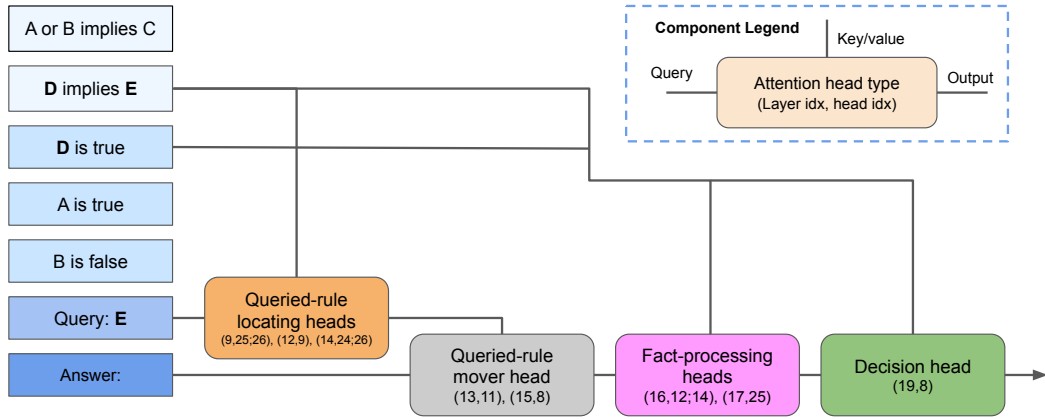

Figure 2: High-level properties of Mistral-7B's reasoning circuit. The (chunks of) input tokens are on the left, which are passed into the residual stream and processed by the attention heads. We illustrate the information flow manipulated by the different types of attention heads we identified to be vital to the reasoning task.

*Observation 3: linearly-decodable linear-chain answer at layer 2.* We train an affine classifier with the same input as the third attention block, with the target being the start of the linear chain; the training samples only query for the linear chain, and we generate 5k of them. We obtain a test accuracy above 97% for this classifier (on 5k test samples), confirming that layer 2 already has the answer at the QUERY position.

**LogOp chain: partial answer in layers 1 & 2 + refinement in layer 3**. To predict the correct starting node of the LogOp chain, the model employs the following strategy:

1. The first two layers encode the LogOp and only a "partial answer". More specifically, we find evidence that (1) when the LogOp is an AND gate, layers 1 and 2 tend to pass the node(s) with FALSE assignment to layer 3, (2) when the LogOp is an OR gate, layers 1 and 2 tend to pass node(s) with TRUE assignment to layer 3.

2. The third layer, combining information of the two starting nodes of the LogOp chain, and the information in the layer-2 residual stream at the ANSWER position, output the correct answer.

We delay the full set of evidence for the above claims to Appendix D.2.

## 4 The reasoning circuit in Mistral-7B

We now turn to examine how a pretrained LLM, namely Mistral-7B solves this reasoning problem. We choose this LLM as it is amongst the smallest accessible model which achieves above 70% accuracy on (a minimal version of) our problem. We present a hypothesis for the reasoning circuit inside the model for predicting the crucial first token of the length-2 problem in Figure 2, and provide evidence relying on a popular technique in mechanistic interpretability, activation patching.

We describe the main properties of the reasoning circuit inside the model for this prediction task in Figure 2. At a high level, there are several intriguing properties of the reasoning circuit of the LLM:[2]

1. Compared to the attention blocks, the MLPs are relatively unimportant to correct prediction.
2. There is a sparse set of attention heads that are found to be central to the reasoning circuit:
    - (Queried-rule locating head) Attention heads (9,25;26), (12,9), (14,24;26) locate the queried rule using the QUERY token, and stores this information at the QUERY position.
    - (Queried-rule mover head) Attention heads (13,11), (15,8) move QUERY and the queried-rule information from the QUERY position to the ":" position.
    - (Fact processing heads) Attention heads (16,12;14), (17,25) locate the relevant facts, and move information to the ":" position.

---

[2]We use $(\ell, h)$ to denote an attention head. When referencing multiple heads in the same layer, we write $(\ell, h_1; h_2; ...; h_n)$ for brevity.

- (Decision head) Attention head (19,8), relying on the aggregated information, makes a decision on which token to output.

## 4.1 Circuit analysis

We only discuss high-level intuitions and results here due to space limitations, and delay the full set of experiments and their interpretations to Appendix E.

Intuitively speaking, to support our hypothesis for the reasoning circuit employed by Mistral-7B to solve the reasoning problem, we rely on activation patching to discover the attention heads which have the greatest influence on the model's output distribution (recall that the MLPs are not as important in this problem). We combine such "causal-mediation" evidence with inspections on these heads' attention patterns. This leads to the set of evidence that is (partially) visualized in Figure 3 below.

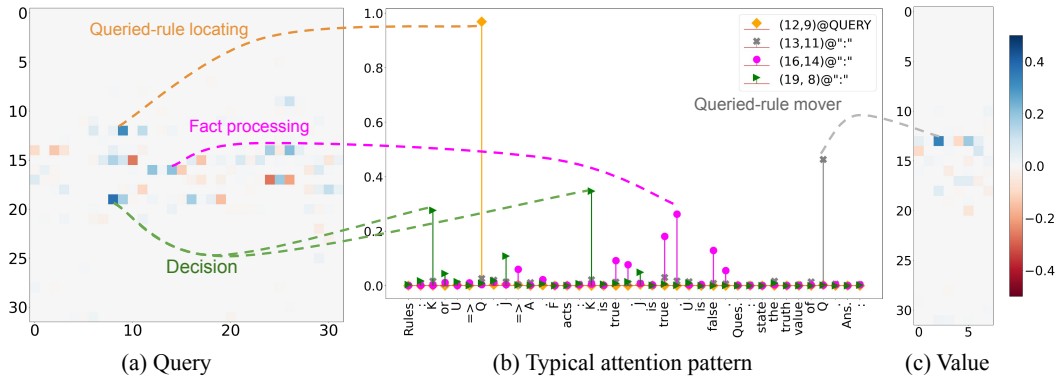

(a) Query        (b) Typical attention pattern        (c) Value

Figure 3: Patching of query and value activations of all attention heads in (a) and (c); we found that intervening the key activations only yield trivial scores, so we do not report them here. We show in (b) the typical attention patterns of a representative set of the attention heads which are identified to be important in the intervention experiments shown in (a) and (c). There are several distinct observations which can be made in (b). Queried-rule locating head (12,9): observe that it correctly locates the queried rule which ends with **Q**. Queried-rule mover head (13,11): the only token position which it focuses on is the QUERY token **Q**. Fact processing head (16,14): attention concentrates in the fact section. Decision head (19,8): attention focused on the correct first answer token **K**.

## 5 Conclusion

We studied the reasoning mechanisms of both small transformers and LLMs on a synthetic propositional logic problem. We analyzed a shallow decoder-only attention-only transformer trained purely on this problem as well as a pretrained Mistral-7B LLM. We uncovered interesting mechanisms the small and large transformers adopt to solve the problem. For the small models, we found the existence of "routing" signals that significantly alter the model's reasoning pathway depending on the sub-category of the problem instance. For Mistral-7B, we found four families of attention heads that implement the reasoning pathway of "QUERY→Relevant Rule→Relevant Facts→Decision". These findings provide valuable insights into the inner workings of LLMs on mathematical reasoning problems.

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

# Appendix / supplemental material

## A  Related works, and scope of this work

**Mechanistic interpretability**. Our work falls in the area of mechanistic interpretability, which aims to understand the mechanisms that enable capabilities of the LLM; such studies involve uncovering certain "circuits" in the network [Elhage et al., 2021, Olsson et al., 2022, Meng et al., 2022, Vig et al., 2020, Feng and Steinhardt, 2024, Wu et al., 2023, Wang et al., 2023, Hanna et al., 2024, Merullo et al., 2024, McGrath et al., 2023, Singh et al., 2024, Feng et al., 2024]. While the definition of a "circuit" varies across different works, in this paper, our definition is similar to the one in Wang et al. [2023]: it is a collection of model components (attention heads, neurons, etc.) with the "edges" in the circuit indicating the information flow between the components in the forward pass; the "excitation" of the circuit is the input tokens.

**Evaluation of reasoning abilities of LLMs**. Our work is also related to the line of work which focus on empirically evaluating the reasoning abilities of LLMs across different types of tasks [Xue et al., 2024, Chen et al., 2024, Patel et al., 2024, Morishita et al., 2023, Seals and Shalin, 2024, Zhang et al., 2023, Saparov and He, 2023, Saparov et al., 2024, Luo et al., 2024, Han et al., 2024, Tafjord et al., 2021, Hendrycks et al., 2021, Dziri et al., 2024, Yang et al., 2024]. While these studies primarily benchmark their performance on sophisticated tasks, our work focuses on understanding "how" transformers reason on logic problems accessible to fine-grained analysis.

**Analysis of how LLMs reason**. There are far fewer studies that focus on providing fine-grained analysis of *how* LLMs reason. To the best of our knowledge, only a handful of works, such as Brinkmann et al. [2024], Xue et al. [2024], Zečević et al. [2023], Ye et al. [2024], share similar goals of understanding how transformers perform multi-step reasoning through detailed empirical or theoretical analysis. However, none studies the [Variable relationships]+[Variable value assignment]+[Query] type problem in conjunction with analysis on *both* small transformers trained purely on the synthetic problem, and large language models trained on a large corpus of internet data.

**Activation patching**. At its core, activation patching, a.k.a. *causal mediation analysis* [Vig et al., 2020, Meng et al., 2022, Hase et al., 2024, Heimersheim and Nanda, 2024, Zhang and Nanda, 2024], uses causal interventions for uncovering the internal mechanisms or "circuits" of LLMs that enable them to perform certain tasks. Typically, the LLM is run on pairs of "source" and "destination" prompts, and we search for components inside the model that "recover" the model's behavior on the source prompts by replacing parts of the model's activation with "source activations" when running on the destination prompt. The opposite "destination→source" intervention can also be adopted.

**Scope of this work**. We define the scope of our analysis as follows. First, in the shallow transformer experiments, we focus on the variant which only has self-attention layers in addition to layer normalization, positional encoding, embedding and softmax parameters. While we could have also included MLP layers, we choose not to because the no-MLP models already achieve 100% accuracy on the problem, and adding MLPs would unnecessarily complicate the analysis. As a second way to focus the scope of paper, in the Mistral-7B experiments, we do *not* seek to uncover *every* model component that participates in solving the reasoning problem. We focus more on finding and analyzing the components that are *necessary* to the model's reasoning circuit, and necessary towards implementing the reasoning pathway as described before. By doing so, we can fully justify the necessity of these key components, without guessing about the roles of less-impactful sub-circuits.

## B  Propositional logic problem and examples

In this section, we provide a more detailed description of the propositional logic problem we study in this paper, and list representative examples of the problem.

At its core, the propositional logic problem requires the reasoner to (1) distinguish which chain type is being queried (LogOp or linear), and (2) if it is the LogOp chain being queried, the reasoner must know what truth value the logic operator outputs based on the two input truth values.

Below we provide a comprehensive list of representative examples of our logic problem at length 2 (i.e. each chain is formed by one rule). We use [Truth values] to denote the relevant input truth value assignments (i.e. relevant facts) to the chain being queried below.

1. Linear chain queried, [True]
   - Rules: A or B implies C. **D implies E**.
   - Facts: A is true. B is true. **D is true**.
   - Question: what is the truth value of C?
   - Answer: D true. D implies E; E True.

2. Linear chain queried, [False]
   - Rules: A or B implies C. **D implies E**.
   - Facts: A is true. B is true. **D is false**.
   - Question: what is the truth value of C?
   - Answer: D false. D implies E; E undetermined.

3. LogOp chain queried, LogOp = OR, [True, True]
   - Rules: A **or** B implies C. D implies E.
   - Facts: **A is true. B is true.** D is true.
   - Question: what is the truth value of C?
   - Answer: B true. A or B implies C; C True.

   **Remark.** *In this case, the answer "A true. A or B implies C; C True" is also correct.*

4. LogOp chain queried, LogOp = OR, [True, False]
   - Rules: A **or** B implies C. D implies E.
   - Facts: **A is true. B is false.** D is true.
   - Question: what is the truth value of C?
   - Answer: A true. A or B imples C; C True.

5. LogOp chain queried, LogOp = OR, [False, False]
   - Rules: A **or** B implies C. D implies E.
   - Facts: **A is false. B is false.** D is true.
   - Question: what is the truth value of C?
   - Answer: A false B false. A or B implies C; C undetermined.

6. LogOp chain queried, LogOp = AND, [True, True]
   - Rules: A **and** B implies C. D implies E.
   - Facts: **A is true. B is true.** D is true.
   - Question: what is the truth value of C?
   - Answer: A true B true. A and B implies C; C True.

7. LogOp chain queried, LogOp = AND, [True, False]
   - Rules: A **and** B implies C. D implies E.
   - Facts: **A is true. B is false.** D is true.
   - Question: what is the truth value of C?
   - Answer: B false. A and B implies C; C undetermined.

8. LogOp chain queried, LogOp = AND, [False, False]
   - Rules: A **and** B implies C. D implies E.
   - Facts: **A is false. B is false.** D is true.
   - Question: what is the truth value of C?
   - Answer: A false. A and B implies C; C undetermined.

   **Remark.** *In this case, the answer "B false. A and B implies C; C undetermined" is also correct.*

The length-3 case is a simple generalization of this set of examples, so we do not cover those examples here.

# C   Learner characteristics, and training details

## C.1   Transformer definition

The architecture definition follows that of GPT-2 closely.

Define input $\boldsymbol{x} = (x_1, x_2, ..., x_T) \in \mathbb{N}^T$, a sequence of tokens with length $T$. It is converted into a sequence of (trainable) token embeddings $\boldsymbol{X}_{token} = (\boldsymbol{e}(x_1), \boldsymbol{e}(x_2), ..., \boldsymbol{e}(x_T)) \in \mathbb{R}^{d_{in} \times T}$. Adding to it the (trainable) positional embeddings $\boldsymbol{P} = (\boldsymbol{p}_1, \boldsymbol{p}_2, ..., \boldsymbol{p}_T) \in \mathbb{R}^{d_{in} \times T}$, we form the zero-th layer embedding of the transformer $\boldsymbol{X}_0 = (\boldsymbol{e}(x_1) + \boldsymbol{p}_1, ..., \boldsymbol{e}(x_T) + \boldsymbol{p}_T)$. The input is processed by the attention blocks as follows.

Let the model have $L$ layers and $H$ heads. For layer index $\ell \in [L]$ and head index $j \in [H]$, attention head $\mathcal{A}_{\ell,j}$ is computed by $\mathcal{A}_{\ell,j}(\boldsymbol{X}_{\ell-1}) = \mathcal{S}\left(\text{causal}\left[\tilde{\boldsymbol{X}}_{\ell-1}\boldsymbol{Q}_{\ell,j}^T\boldsymbol{K}_{\ell,j}\tilde{\boldsymbol{X}}_{\ell-1}^T\right]/\sqrt{d_H}\right)\tilde{\boldsymbol{X}}_{\ell-1}\boldsymbol{V}_\ell^T$, with $\tilde{\boldsymbol{X}}_{\ell-1} = \text{LayerNorm}(\boldsymbol{X}_{\ell-1})$, $\mathcal{S}(\cdot)$ being the softmax operator, causal$[\cdot]$ the causal mask operator. The output of the attention block is $\mathcal{A}_\ell = \boldsymbol{X}_{\ell-1} + \text{Concat}[\mathcal{A}_{\ell,1}(\boldsymbol{X}_{\ell-1}), ..., \mathcal{A}_{\ell,H}(\boldsymbol{X}_{\ell-1})]\boldsymbol{W}_{O,\ell}^T$, with $\boldsymbol{W}_{O,\ell}$ the square projection matrix (with bias). Finally, we apply an affine classifier (with softmax) $\boldsymbol{f}(\boldsymbol{x}) = \mathcal{S}(\tilde{\boldsymbol{X}}_{L,T}\boldsymbol{W}_{class}^T + \boldsymbol{b}_{class})$ to predict the next word.

In this paper, we set the hidden space embedding to 768.

## C.2   Training details

In all of our experiments, we set the learning rate to $10^{-4}$, and weight decay to $10^{-4}$. For models with depth less than 6, we use a batch size of 512, and train the model for 60k iterations; for models with depth greater than or equal to 6, we use a batch size of 256, and train for 80k iterations. We use the AdamW optimizer in PyTorch, with 5k iterations of linear warmup, following by cosine annealing to a learning rate of 0. Each model is trained on a single V100 GPU; the full set of models take around 2 - 3 days to finish training.

# D   Section 3 Experimental setup and observations

**Problem specification**. In each logic problem instance, the proposition variables are randomly sampled from a pool of 80 variables (tokens). The truth values in the fact section are also randomly chosen. The linear chain is queried 20% of the time; the LogOp chain is queried 80% of the time.

**Accuracy of different variants of the model for the length-3 problem**. We show in Figure 4 below that the 3-layer 3-head variant is the smallest model which achieves $\tilde{1}00\%$ accuracy on the problem.

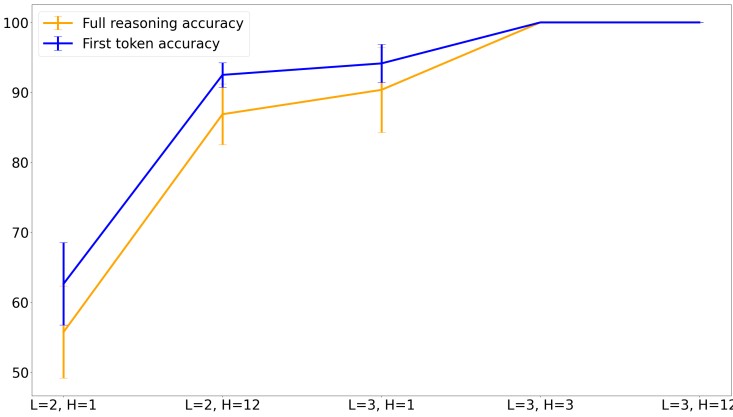

Figure 4: Accuracy of full reasoning and first token for several models on the length-3 problem.

### D.1 Routing signal in the second attention block

**Observation 1: chain-type disentanglement at QUERY**. We construct 200 samples, with the first half querying the linear chain, the second half querying the logical-operator chain. We record the second-layer's self-attention block output on these samples, and compute the cosine similarity between each pair. We show the result in Figure 5.

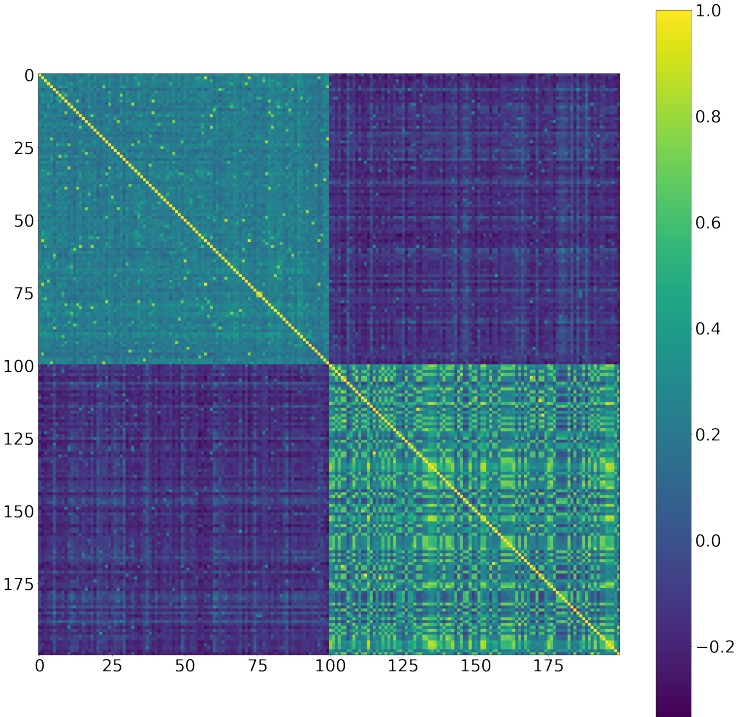

Figure 5: Disentanglement based on whether QUERY is for the linear chain, observed at the second self-attention block.

**Observation 2a: existence of an abstract "routing signal"**. We make the following experimental observations.

1. We generate 1,000 samples whose QUERY is for the linear chain, and compute the average output embedding of layer-2 self-attention block (post projection matrix). We denote this embedding as $h_{route}$.

2. (Linear→LogOp intervention) Sample a set of 500 validation samples, all of which query for the *linear chain*. In the forward pass of the model on every validation sample, we *subtract* the $h_{route}$ to the output of the second attention block — note that this "corrupted" signal from the second layer is also received by the third layer. We observe that the model's first token prediction is 100% of the time the correct first token for the LogOp chain.

3. (LogOp→linear intervention) We repeat the above experiment the other way around. We sample another 500 validation samples, but in this case they all query the LogOp chain. On every sample, during the forward pass we *add* $h_{route}$ to the output of the second attention block. We record the attention weight of the third layer's attention blocks at the ANSWER token position. We then average these attention weights for each head. We find that all three attention heads place greater than 99% attention weight on the QUERY position on average, behaving exactly like when the sample naturally queries for the linear chain.

**Observation 2b: linearly-decodable linear-chain answer at layer 2**. We simply frame the learning problem as a linear classification problem. The input vector of the classifier is the same as the input to the layer-3 self-attention block, equivalently the layer-2 residual-stream embedding. The output space is the set of proposition variables (80-dimensional vector). We train the classifier on 5k training samples (all whose QUERY is for the linear chain) using the AdamW optimizer, with learning rate

set to $5 \times 10^{-3}$ and weight decay of $10^{-2}$. We verify that the trained classifier obtains an accuracy greater than 97% on an independently sampled test set of size 5k (all whose QUERY is for the linear chain too).

## D.2 Answer for the LogOp Chain

*Evidence 3a: Distinct behaviors of affine predictors at different layers.* We train two affine classifiers at two positions inside the model (each with 10k samples): $\boldsymbol{W}_{resid,\ell=2}$ at layer-2 residual stream, and $\boldsymbol{W}_{attn,\ell=3}$ at layer-3 attention-block output, both at the position of ANSWER, with the target being the correct first token. In training, if there are two correct answers possible (e.g. OR gate, starting nodes are both TRUE or both FALSE), we randomly choose one as the target; in testing, we deem the top-1 prediction "correct" if it coincides with one of the answers. We observe the following predictor behavior on the test samples:

1. $\boldsymbol{W}_{attn,\ell=3}$ predicts the correct answer 100% of the time.
2. $\boldsymbol{W}_{resid,\ell=2}$ always predicts one of the variables assigned FALSE (in the fact section) if LogOp is the AND gate, and predicts one assigned TRUE if LogOp is the OR gate.

*Evidence 3b: linearly decodable LogOp information from first two layers.* We train an affine classifier at the layer-2 residual stream to predict the LogOp of the problem instance, over 5k samples (and tested on another 5k samples). The classifier achieves greater than 98% accuracy. We note that training this classifier at the layer-1 residual stream also yields above 95% accuracy.

*Evidence 3c: identification of LogOp-chain starting nodes at layer 3.* Attention heads (3,1) and (3,3), when concatenated, produce embeddings which we can linearly decode the two starting nodes of the LogOp chain with test accuracy greater than 98%. We also find that they focus their attention in the rule section of the context (as shown in Figure 6). Due to causal attention, this means that they determine the two starting nodes from the LogOp-relevant rules.

*Remark.* The above pieces of observations suggest the "partial information→refinement" process.[3] To further validate that the embedding from the first two layers are indeed causally linked to the correct answer at the third layer, we perform an activation patching experiment.

*Evidence 3d: layer-2 residual stream at ANSWER is important to correct prediction.* We verify that layer-3 attention does rely on information in the layer-2 residual stream (at the ANSWER position):

- Construct two sets of samples $\mathcal{D}_1$ and $\mathcal{D}_2$, each of size 10k: for every sample $\boldsymbol{X}_{1,n} \in \mathcal{D}_1$ and $\boldsymbol{X}_{2,n} \in \mathcal{D}_2$, the context of the two samples are exactly the same, except the LogOp is flipped, i.e. if $\boldsymbol{X}_{1,n}$ has disjunction, then $\boldsymbol{X}_{2,n}$ has the conjunction operator. If layer 3 of the model has *no* reliance on the $\text{Resid}_{\ell=2}$ (layer-2 residual stream) for LogOp information at the ANSWER position, then when we run the model on any $\boldsymbol{X}_{2,n}$, patching $\text{Resid}_{\ell=2}(\boldsymbol{X}_{n,2})$ with $\text{Resid}_{\ell=2}(\boldsymbol{X}_{n,1})$ at ANSWER should *not* cause significant change to the model's accuracy of prediction. However, we observe the contrary: the accuracy of prediction degrades from 100% to 70.87%, with standard deviation 3.91% (repeated over 3 sets of experiments).

*Observation: LogOp-relevant reasoning at the third layer.* We show that the output from attention heads (3,1) and (3,3) (before the output/projection matrix of the layer-3 attention block), namely $\mathcal{A}_{3,1}(\boldsymbol{X}_2)$ and $\mathcal{A}_{3,3}(\boldsymbol{X}_2)$, when concatenated, contain linearly decodable information about the two starting nodes of the LogOp chain. We frame this as a multi-label classification problem, detailed as follows:

1. We generate 5k training samples and 5k test samples, each of whose QUERY is for the LogOp chain. For every sample, we record the *target* as a 80-dimension vector, with every entry set to 0 except for the two indices corresponding to the two proposition variables which are the starting nodes of the LogOp chain.

2. Instead of placing softmax on the final classifier of the transformer, we use the Sigmoid function. Moreover, instead of the Cross-Entropy loss, we use the Binary Cross-Entropy loss (namely the

---

[3]In fact, the observations suggest that layer 3 performs a certain "matching" operation. Take the OR gate as an example. Knowing which of the three starting nodes (for LogOp and linear chain) are TRUE, and which two nodes are the starting nodes for the LogOp chain are sufficient to determine the first token! This exact algorithm, however, is not fully validated by our evidence; we leave this as part of our future work.

`torch.nn.functional.binary_cross_entropy_with_logits` in PyTorch, which directly includes the Sigmoid for numerical stability).

3. We train an affine classifier, with its input being the concatenated $\text{Concat}[\mathcal{A}_{3,1}(\boldsymbol{X}_2), \mathcal{A}_{3,3}(\boldsymbol{X}_2)]$ (a 512-dimensional vector) on every training sample, and with the targets and training loss defined above. We use a constant learning rate of $0.5 \times 10^{-3}$, and weight decay of $10^{-2}$. The optimizer is AdamW in PyTorch.

4. We assign a "correct" evaluation of the model on a test sample only if it correctly outputs the two target proposition variable as the top-2 entries in its logits. We observe that the classifier achieves greater than 98% once it converges.

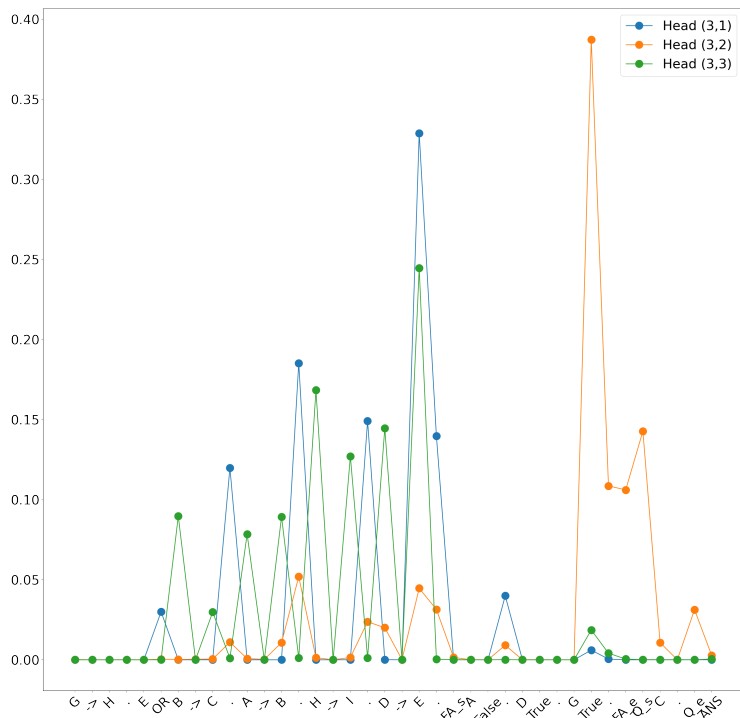

Figure 6: Attention statistics, averaged over 500 samples, all of which query for the LogOp chain. The x-axis is simply an example prompt that helps illustrate where the attention is really placed at. Observe that only attention head (3,2) pays significant attention to the fact section. The other two heads focus on the rule section.
Reminder: due the the design of the problem, the rule, fact and query sections all have consistent length for every sample!

### D.3 Extra remarks

**Observation 3 supplement: linearly-decodable linear-chain answer at layer 2**. We simply frame the learning problem as a linear classification problem. The input vector of the classifier is the same as the input to the layer-3 self-attention block, equivalently the layer-2 residual-stream embedding. The output space is the set of proposition variables (80-dimensional vector). We train the classifier on 5k training samples (all whose QUERY is for the linear chain) using the AdamW optimizer, with learning rate set to $5 \times 10^{-3}$ and weight decay of $10^{-2}$. We verify that the trained classifier obtains an accuracy greater than 97% on an independently sampled test set of size 5k (all whose QUERY is for the linear chain too).

**Remarks on truth value determination**. Evidence suggests that determining the truth value of the simple propositional logic problem is easy for the model, as the truth value of the final answer is linearly decodable from layer-2 residual stream (with 100% test accuracy, trained on 10k samples) when we give the model the context+chain of thought right before the final truth value token. This is expected, as the main challenge of this logic problem is not about determining the query's truth

value, but about the model spelling out the minimal proof with careful planning. When abundant CoT tokens are available, it is natural that the model knows the answer even in its second layer.

# E  Mistral 7B: Experimental Details

## E.1  Problem format

In our Mistral-7B experiments, the input samples have the following properties:

1. We give the model 6 (randomly chosen) in-context examples before asking for the answer.

2. The problem is length-2: only one rule involving the OR gate, and one linear-chain rule. Moreover, the answer is always true. In particular, the truth values of the two premise nodes of the OR chain always have one FALSE and one TRUE.

3. The proposition variables are all (single-token) capital English letters.

The design decision in the first point is to ensure fairness to the LLM which was not trained on our specific logic problem. As for the last two point, we restrict the problem in this fashion mainly to ensure that the first answer token is *unique*, which improves the tractability of the analysis. Note that these restrictions do not take away the core challenge of this problem: *the LLM still needs to process all the context information without CoT to determine the correct first token.*

An example problem is presented below.

```
Rules: Z or F implies B. D implies C.
Facts: D is true. Z is true. F is false.
Question: state the truth value of C.
Answer: D is true. D implies C; C is true.
Rules: U implies Y. G or I implies Q.
Facts: I is true. U is true. G is false.
Question: state the truth value of Y.
Answer: U is true. U implies Y; Y is true.
Rules: G or Z implies E. U implies K.
Facts: U is true. G is true. Z is false.
Question: state the truth value of E.
Answer: G is true. G or Z implies E; E is true.
Rules: G implies U. Y or A implies V.
Facts: Y is true. G is true. A is false.
Question: state the truth value of V.
Answer: Y is true. Y or A implies V; V is true.
Rules: U implies W. H or B implies L.
Facts: B is false. U is true. H is true.
Question: state the truth value of W.
Answer: U is true. U implies W; W is true.
Rules: F or A implies Y. E implies I.
Facts: A is false. F is true. E is false.
Question: state the truth value of Y.
Answer: F is true. F or A implies Y; Y is true.
Rules: B or F implies D. S implies T.
Facts: S is true. F is true. B is false.
Question: state the truth value of T.
Answer:
```

**Remark.** *To ensure fairness to the LLM, we balance the number of in-context examples which queries the OR chain and the linear chain: each has 3 in-context examples. The order in which the in-context examples are presented (i.e. the order in which the examples with OR or linear-chain answer) is random.*

## E.2 Causal mediation analysis

We provide evidence in this part of the paper primarily relying on a popular technique in mechanistic interpretability: *causal mediation analysis*. Our methodology is roughly as follows:

1. Suppose we are interested in the role of the activations of certain components of the LLM in a certain (sub-)task. For a running example, say we want to understand what role the attention heads play in processing and passing QUERY information to the ":" position for inference. Let us denote the activations as $\boldsymbol{A}_{\ell,h;t}(\boldsymbol{X})$, representing the activation of head $h$ in layer $\ell$, at token position $t$.

2. Typically, the analysis begins by constructing two sets of prompts which differ in subtle ways. A natural construction in our example is as follows: define sets of samples $\mathcal{D}_{orig}$ and $\mathcal{D}_{alt}$, where $\boldsymbol{X}_{orig,n}$ and $\boldsymbol{X}_{alt,n}$ have exactly the same context, except in $\boldsymbol{X}_{orig,n}$, QUERY is for the LogOp chain, while in $\boldsymbol{X}_{alt,n}$, QUERY is for the linear chain. Moreover, denote the correct targets $y_{orig,n}$ and $y_{alt,n}$ respectively.

3. We run the LLM on $\mathcal{D}_{orig}$ and $\mathcal{D}_{alt}$, caching the attention-head activations. We also obtain the logits of the model. We can compute the model's *logit differences*

$$\Delta_{orig,n} = \text{logit}(\boldsymbol{X}_{orig,n})[y_{orig,n}] - \text{logit}(\boldsymbol{X}_{orig,n})[y_{alt,n}].$$

   For a high-accuracy model, $\Delta_{orig,n}$ should be *large* for most $n$'s, since it must be able to clearly tell that on an $\boldsymbol{X}_{orig,n}$, it is the LogOp chain which is being queried, not the linear chain.

4. We now perform intervention for all $n, \ell, h$ and $t$:
   (a) Run the model on $\boldsymbol{X}_{orig,n}$, however, replacing the original activation $\boldsymbol{A}_{\ell,h;t}(\boldsymbol{X}_{orig,n})$ by the altered $\boldsymbol{A}_{\ell,h;t}(\boldsymbol{X}_{alt,n})$. Now let the rest of the run continue.[4] Let us denote the logits obtained in this intervened run as $\text{logit}^{\rightarrow alt;(\ell,h,t)}(\boldsymbol{X}_{orig,n})$.
   (b) Now compute the intervened logit difference

$$\Delta_{orig \rightarrow alt,n;(\ell,h,t)} = \text{logit}^{\rightarrow alt;(\ell,h,t)}(\boldsymbol{X}_{orig,n})[y_{alt,n}] - \text{logit}^{\rightarrow alt;(\ell,h,t)}(\boldsymbol{X}_{orig,n})[y_{orig,n}].$$

5. Now average the $\Delta_{orig \rightarrow alt,n;(\ell,t)}$'s over $n$ for every $\ell, h$ and $t$ (recall that $n$ is the sample index).

6. This procedure helps us identify components that are significant in processing and passing the QUERY information for inference. Intuitively, an activation that result in a positive and large $\Delta_{orig \rightarrow alt,n;(\ell,t)}$ play a significant role in this subtask, because this activation helps "altering" the model's "belief" from "QUERY is for the LogOp chain" to "QUERY is for the linear chain".

7. *Remark*: due to the symmetry of this running example, it is perfectly sensible to perform $alt \rightarrow orig$ interventions too, by mirroring the above procedures.

Each of our experiments are done on 60 samples unless otherwise specified — we repeat some experiments (especially the attention-head patching experiments) to ensure statistical significance when necessary.

**Calibrated metric**. Please note that in this work, we adopt a calibrated/normalized version of the intervened logit difference (aimed at keeping the score's magnitude between 0 and 1). In particular, we compute the following metric for head $(\ell, h)$ at token position $t$:

$$\frac{\frac{1}{N} \sum_{n \in [N]} \Delta_{orig \rightarrow alt,n;(\ell,h,t)} - \Delta_{orig}}{\Delta_{alt} - \Delta_{orig}}. \tag{1}$$

where $\Delta_{orig} = \frac{1}{N} \sum_{n \in [N]} \text{logit}(\boldsymbol{X}_n)[y_{alt,n}] - \text{logit}(\boldsymbol{X}_n)[y_{orig,n}]$, and $\Delta_{alt} = \frac{1}{N} \sum_{n \in [N]} \text{logit}(\boldsymbol{X}'_n)[y_{alt,n}] - \text{logit}(\boldsymbol{X}'_n)[y_{orig,n}]$. The closer to 1 this score is, the stronger the model's belief is altered.

## E.3 QUERY-based patching: discovering the important attention heads

Our analysis relies on QUERY-based patching, following the same procedure as detailed in subsection E.2. In this set of experiments, we discover the main attention heads responsible for processing the context and performing inference as introduced in the beginning of this Section.

---

[4]Please note that layers $\ell + 1$ to $L$ are influenced at and after token position $t$, and technically speaking, now operate "out of distribution".

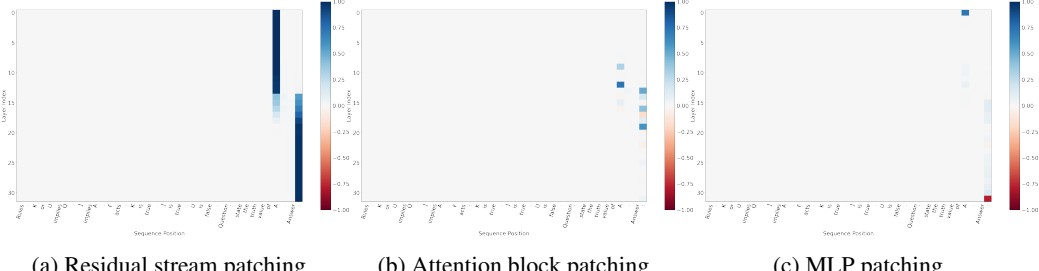

| (a) Residual stream patching | (b) Attention block patching | (c) MLP patching |

Figure 7: Query patching at the level of residual streams, attention blocks and MLPs. Highly localized processing of QUERY is observed: a sharp transition occurs at layer 13 in (a), and in (b), only a sparse set of attention blocks play a major role in this subtask. Furthermore, (c) shows that the MLPs play a limited role in this subtask (besides MLP0). (Please zoom in for the details)

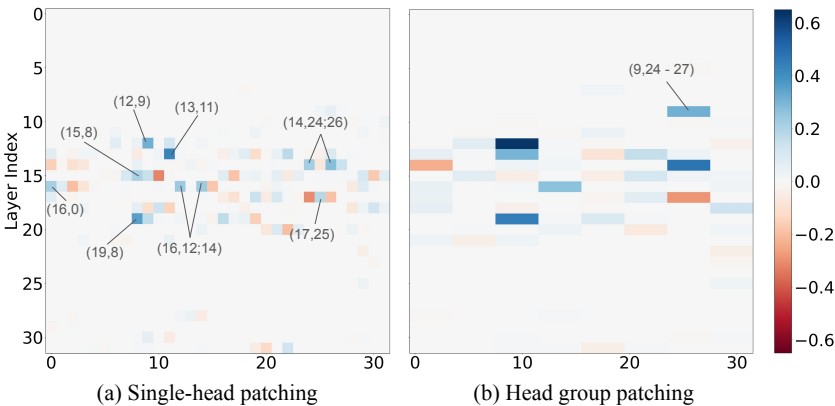

| (a) Single-head patching | (b) Head group patching |

Figure 8: Attention head patching, highlighting the ones with the highest intervened logit difference; $x$-axis is the head index. (a) shows single-head patching, and (b) shows a coarser-grained head patching in groups. In (b), we only highlight the head groups that are not captured well by (a).

**Why is QUERY-based patching important to reasoning circuit discovery?** To answer this question, there are two points to emphasize first. (1) We know that to solve the reasoning problem, the QUERY token is critical to *initiating* the reasoning chain: without it, the rules and facts are completely useless; with it, the reasoner can then proceed to identify the relevant rules and facts to predict the answer. (2) The prompt pairs differ *only* by the QUERY token. Based on (1) and (2), we know that if performing the aforementioned QUERY-based causal intervention on a model component leads to a large intervened logit difference (i.e. it alters the model's "belief"), then this component must be integral to the reasoning circuit, because the component is now identified to be QUERY-sensitive *and* has causal influence on (parts of) the model's reasoning actions.

*High-level interventions.* We begin by presenting higher-level patching results in Figure 7, where we intervene at the level of residual streams, attention blocks, and MLPs. We can draw a few insights from these results:

1. A sharp transition of "QUERY processing" occurs from layer 12 to layer 13 (indexed from 0) in Figure 7(a) and (b).

2. Figure 7(b) shows that a small set of attention blocks are observed to be significant for the "belief altering" action, namely those in layers 9, 12, 13, 14, 16 and 19.

3. The MLPs, shown in Figure 7(c), play little role in this circuit, except for MLP-0. However, MLP-0 had been observed to act more as a "nonlinear token embedding" than a complex high-level processing unit [Wang et al., 2023]. In the rest of this section, we primarily devote our analysis to the attention heads, and leave the exact role of the MLPs to future work.

*Attention-head interventions.* Figure 7 helps us locate a small set of attention blocks which are important to the task. However, these results alone still leave us with far too many components to

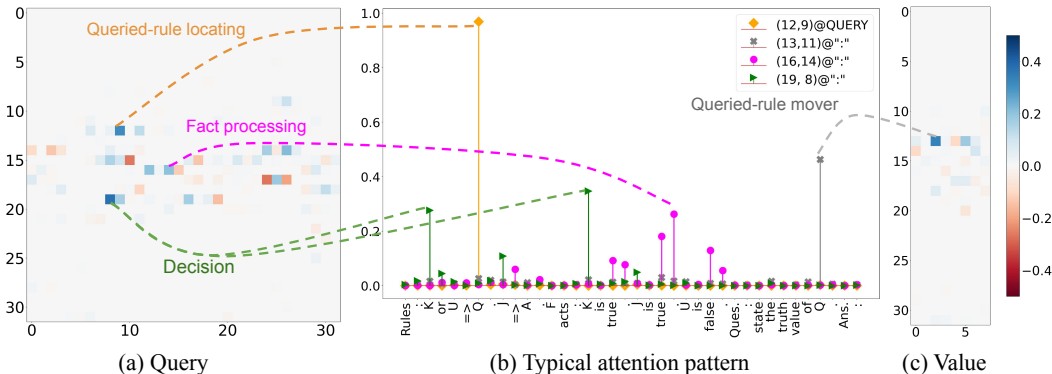

|  |  |  |
|---|---|---|
| (a) Query | (b) Typical attention pattern | (c) Value |

Figure 9: Patching of query and value activations in (a) and (c); we found that intervening the key activations only yield trivial scores, so we do not report them here. We show in (b) the typical attention patterns of a representative set of the attention heads which are identified to be important in the intervention experiments shown in (a) and (c). There are several distinct observations which can be made in (b). Queried-rule locating head (12,9): observe that it correctly locates the queried rule which ends with **Q**. Queried-rule mover head (13,11): the only token position which it focuses on is the QUERY token **Q**. Fact processing head (16,14): attention concentrates in the fact section. Decision head (19,8): attention focused on the correct first answer token **K**.

examine in detail. Therefore, we run a set of finer-grained experiments, intervening on the attention heads (over the relevant context). The results are shown in Figure 8. We find that, interestingly, only a very small set of attention heads are central to the "belief altering" of the LLM. More specifically, in Figure 8(a), only attention heads (12,9), (13,11), (14,24;26), (16,0;12;14), (17,25), (19,8) are observed with relatively high intervened logit differences.

We note that *Grouped-Query Attention* used by Mistral-7B adds subtlety to the analysis[5]: patching a single head might not yield a high logit difference since other heads in the same group (which possibly perform a similar function) could overwhelm the patched head and maintain the model's previous "belief". Therefore, we also run a *coarser-grained* experiment which simultaneously patches the attention heads sharing the same key and value activations, shown in Figure 8(b). This experiment reveals that heads belonging to the group (9, 24 - 27) also have high intervened logit difference.

### E.4 Causal interventions on the sub-components of attention heads

We aim to understand why the attention heads identified in the last sub-section are important. For now, we continue with QUERY altering in the prompt pairs. Through intervening on the sub-components of each attention head, namely their value, key, and query, and through examining details of their attention weights, we find that there are roughly four types of attention heads. We show the results in Figure 9 (repeated here in the Appendix for convenience):

1. Queried-rule locating head. Attention head (12,9)'s *query* activation has a large intervened logit difference according to Figure 3(a), therefore, its query and attention patterns are QUERY-dependent and contribute to altering the model's "belief". Furthermore, at the QUERY position, we find that *on average*, its attention weight is above 90% at the "conclusion" variable of the rule being queried. In other words, it is responsible for *locating* the queried rule, and storing that rule's information at the QUERY position.[6]

2. Queried-rule mover head. Attention head (13,11)'s *value* activations have large intervened logit difference, and intriguingly, its query and key activations do *not* share that tendency. This already suggests that its attention pattern performs a fixed action on both the original and altered prompts, and only the value information is sensitive to QUERY. Furthermore, within the relevant context (excluding the 6 in-context examples given), head (13,11) assigns above 50% attention weight to the QUERY position, and its attention weight at QUERY is about 10 times larger than the second

---

[5]In Mistral-7B-v0.1, each attention layer has 8 key and value activations, and 32 query activations. Therefore, heads $(\ell, h \times 4)$ to $(\ell, h \times 4 + 3)$ share the same key and value activation.

[6]Heads (9,25;26), (14,24;26) exhibit similar tendencies, albeit with smaller intervened logit differences.

largest one on average. Recalling the role of layer 12, we find evidence that head (13,11) moves the QUERY and queried-rule information to the ":" position.[7]

3. Fact processing heads. Attention heads (16,12), (16,14) and (17,25)'s *query* activations have large intervened logit differences. Within the relevant context, they place greater than 56%, and 70% of their attention respectively in the fact section of the context (starting from "Fact" and ending on "." before "Question").

4. Decision head. Attention head (19,8)'s query activations have large intervened logit differences. Its attention pattern suggests that it is a "decision" head: within the relevant context, when the model is *correct*, the head's top-2 attention weights are always on the correct starting node of the queried rule and the correct variable in the fact section, and the two token positions occupy more than 60% of its total attention in the relevant context on average. In other words, it already has the answer.

### E.5 Attention patterns of QUERY-sensitive attention heads

In this subsection, we provide finer details on the attention patterns of the attention heads we discovered in Section E.3. Note that the attention weights percentage we present in this section are calculated by dividing the observed attention weight at a token position by the total amount of attention the head places in the relevant context, i.e. the portion of the prompt which excludes the 6 in-context examples.

**Queried-rule locating heads**. Figure 10 presents the average attention weight the queried-rule locating heads place on the "conclusion" variable and the period "." immediately after the queried rule at the QUERY token position (i.e. the query activation of the heads come from the residual stream at the QUERY token position) — (12,9) is an exception to this recording method, where we only record its weight on the conclusion variables alone, and already observe very high weight on average. The heads (12,9), (14,24), (14,26), (9,25), (9,26) indeed place the majority of their attention on the correct position *consistently* across the test samples. The reason for counting the period after the correct conclusion variable as "correctly" locating the rule is that, it is known that LLMs tend to use certain "register tokens" to record information in the preceding sentence.

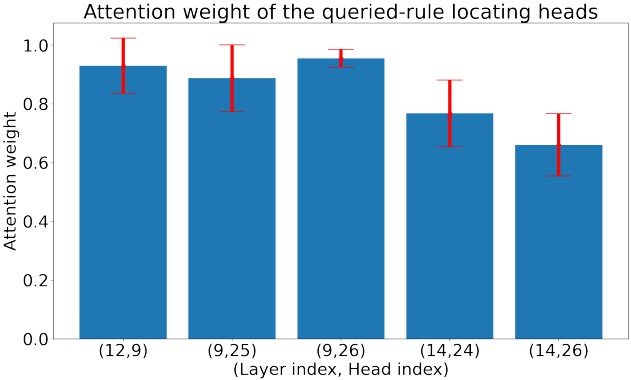

Figure 10: Average attention weights of the queried-rule locating heads, along with the standard deviations. The weights are calculated by dividing the actual attention weight placed on the correct "conclusion" variable of the rule and the period "." immediately after, by the total amount of attention placed in the relevant context (i.e. the prompt excluding the 6 in-context examples). *Head (12,9) is an exception: we only record its attention right on the conclusion variable, and still observe $93.0 \pm 9.4\%$ "correctly placed" attention on average.*

We can observe that head (12,9) has the "cleanest" attention pattern out of the ones identified, placing on average $93.0 \pm 9.4\%$ of it attention on the correct conclusion variable alone. The more diluted attention patterns of the other heads likely contribute to their weaker intervened logit difference score shown in Section E.3 in the main text.

---

[7]Heads (15,8), (16,0) also appear to belong to this type, albeit with smaller intervened logit difference.

**Queried-rule mover heads**. Figure 11 shows the attention weight of the queried-rule mover heads. While they do not place close to 100% attention on the QUERY location consistently (when the query activation comes from the residual stream from token ":", right before the first answer token), the top-1 attention weight consistently falls on the QUERY position, and the second largest attention weight is much smaller. In particular, head (13,11) places $54.2 \pm 12.5\%$ attention on the QUERY position on average, while the second largest attention weight in the relevant context is $5.2 \pm 1.1\%$ on average (around 10 times smaller; *this ratio is computed per sample and then averaged*).

An extra note about head (16,0): it does *not* primarily act like a "mover" head, as its attention statistics suggest that it processes the *mixture* of information from the QUERY position and the ":" position. Therefore, while we present its statistics along with the other queried-rule mover heads here since it does allocate significant attention weight on the QUERY position on average, we do not list it as such in the circuit diagram of Figure 2.

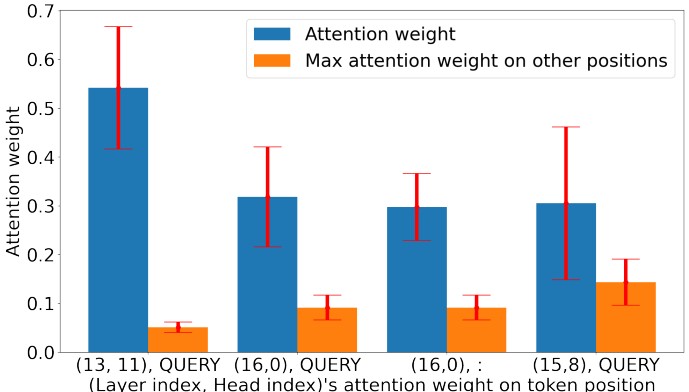

Figure 11: Average attention weights of the queried-rule mover heads, along with the standard deviations. The raw attention patterns are obtained at token position ":" (i.e. the query activation comes from the residual stream at the ":" position), right before the first answer token, and the exact attention weight (indicated by the blue bars) is taken at the QUERY position; for head (16,0), we also obtain its attention weight at the ":" position, as we found that it also allocates a large amount of attention weight to this position in addition to the QUERY position. Note: for (15,8), we found that it only acts as a "mover" head when the linear chain is being queried, so we are only reporting its attention weight statistics in this specific scenario; the other heads do not exhibit this interesting behavior, so we report those heads' statistics in all query scenarios.

**Fact processing heads**. Figure 12 below shows the attention weights of the fact processing heads; the attention patterns are obtained at the ":" position, right before the first answer token, and we sum the attention weights in the Fact section (starting at the first fact assignment, ending on the last "." in this section of the prompt). It is clear that they place significant attention on the Fact section of the relevant context.

**Remark.** *There is only one "decision head" which we identified, i.e. head (19,8). Since there is no further subtleties with how we recorded its attention weights or peculiar behaviors of the attention patterns observed, we do not elaborate further on it in the Appendix.*

# F   Potential Broader Impact

This paper presents work whose goal is to advance the field of Machine Learning, particularly the area of mechanistic interpretability. There are many potential societal consequences of our work, none which we feel must be specifically highlighted here

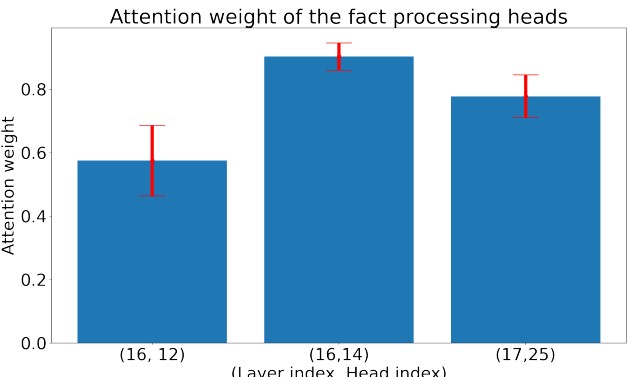

Figure 12: Average attention weights of the fact processing heads, along with the standard deviations. The weights are calculated by dividing the actual attention weight placed in the Fact section by the total amount of attention placed in the relevant context (i.e. the part of the prompt excluding the 6 in-context examples).

