# OpenReview forum: "How Transformers Reason: A Case Study on a Synthetic Propositional Logic Problem"
_NeurIPS.cc/2024/Workshop/MATH-AI — MATH-AI 24_

### Official Review · Reviewer_FLPQ · 2024-10-01

**Rating:** 6
**Confidence:** 2

**Review:**

This paper presents a study on a synthetic propositional logic problem to show whether Transformers can reason and they adopt a three-layer Transformer to decipher this mechanism.

The main result of this manuscript is:
*They discover that small transformers trained purely on the synthetic problem repurposes certain tokens of the context to filter out information that are less relevant to the answer.*


After reading this paper, I find that this paper is hard to follow for the following reasons:
1. The organization of this paper is bad, the results of this paper are not easy to understand.
2. I tried to catch the main idea but could not understand what this paper did.
For example, in section 2.1:

> For example, as shown in Figure 1, only the fact "D False" ...

However, no `D` is shown in Figure 1.

> if QUERY is for the LO chain ...

So what is LO chain? Is it different from the linear chain? Why are they different?

**Pros**
- The problem studied is very interesting, and the use of propositional logic is fundamental and potentially a good choice.

**Cons**
- Poor writing and organization hinder the reader's understanding of this paper.
- The experiments seem interesting but are not convincing since the authors only conducted them on a very specific case.

Overall, I find this paper interesting but it needs improvement in organization and detail. It would be helpful if the authors clarified their main claims and improved the writing.

---

### Official Review · Reviewer_xbox · 2024-10-08

**Rating:** 7
**Confidence:** 4

**Review:**

**Summary**:
This paper investigates how transformers solve a synthetic propositional logic task, providing a detailed analysis of the mechanisms by which a 3-layer transformer and the larger Mistral-7B model process reasoning tasks. The authors identify key attention circuits responsible for reasoning, showing how transformers use specific layers and attention heads to handle different types of queries (linear vs. logical operator chains). The study uses activation patching to analyze Mistral-7B's reasoning processes.

**Strengths**:
- **Problem Selection**: The synthetic logic problem is a well-chosen, controlled task for understanding reasoning in transformers.
- **In-Depth Analysis**: The paper offers rigorous experimentation, particularly with activation patching, providing deep insights into transformer architectures.
- **Interesting Findings**: The discovery of "routing signals" and the analysis of Mistral-7B's layer-specific reasoning are valuable contributions.

**Weaknesses**:
- **Limited Task Scope**: The study focuses on only one synthetic propositional logic problem, limiting the generalizability of the findings to other reasoning domains or tasks.

---

### Official Review · Reviewer_Ma3e · 2024-10-08
**Review of How Transformers Reason: A Case Study on a Synthetic Propositional Logic Problem**

**Rating:** 6
**Confidence:** 1

**Review:**

Summary:
This paper explores how transformers reason through a synthetic propositional logic task, focusing on specific mechanisms in a small 3-layer model and the mistral-7b model. It identifies key patterns, such as query-type disentanglement and reasoning circuits, using methods like activation patching.

Strengths:
- mechanistic insight: the detailed analysis of attention heads and routing signals provides valuable insights into how transformers handle reasoning tasks.
- empirical results: the observations around ''planning'' and ''reasoning'' circuits, especially the disentanglement at the query level, are interesting and well supported by experiments.
- good use of interpretability methods: the use of activation patching in larger models like mistral-7b is a nice point.

Suggestions:
- connect small and large models: it would help to clarify how findings from small transformers extend to larger models.

- expand literature review: the current review feels limited. i would strongly suggest the authors to expand it, providing readers with a more comprehensive understanding of existing works and contexts in mechanistic interpretability, and implications on future works.

- explore robustness of routing signals: the routing signals in the second attention block are interesting. additional experiments testing how these signals behave with different logic tasks or architectures could offer more insight into their general applicability.


Please note that i am not a domain expert in this specific topic and its related literature, but i found the paper’s explanations clear and the results convincing in general, thus i lean towards acceptance.

---

### Decision · Program_Chairs · 2024-10-08

**Decision:**

Accept

**Comment:**

The paper provides an interesting mechanistic analysis of how transformers reason. The authors are suggested to improve the clarity of the paper and try out some other domains to enhance the soundness of their work.